# Retrospective Evaluation of Clinical and Clinicopathologic Findings, Case Management, and Outcome for Dogs and Cats Exposed to *Micrurus fulvius* (Eastern Coral Snake): 92 Cases (2021–2022)

**DOI:** 10.3390/toxins16060246

**Published:** 2024-05-27

**Authors:** Jordan M. Sullivan, Taelor L. Aasen, Corey J. Fisher, Michael Schaer

**Affiliations:** Department of Small Animal Clinical Sciences, Veterinary Teaching Hospital, University of Florida, 2015 S.W. 16th Ave, Gainesville, FL 32610-0144, USA; jsullivan18@ufl.edu (J.M.S.); aasendvm@gmail.com (T.L.A.); cfisher25@ufl.edu (C.J.F.)

**Keywords:** coral snake, dogs, cats, antivenom, neurotoxin, hemolysis

## Abstract

This retrospective, observational study describes the clinical findings, case management trends, and outcomes of 83 dogs and nine cats exposed to eastern coral snakes in a university teaching hospital setting. The medical records of dogs and cats that received antivenom following coral snake exposure were reviewed. Data collected included signalment, time to antivenom administration, physical and laboratory characteristics at presentation, clinical course during hospitalization, length of hospitalization, and survival to discharge. The mean time from presentation to coral snake antivenom administration was 2.26 ± 1.46 h. Excluding cases where the owner declined in-hospital care, the mean hospitalization time for dogs and cats was 50.8 h and 34 h, respectively. The mean number of antivenom vials was 1.29 (1–4). Gastrointestinal signs (vomiting and ptyalism) occurred in 42.2% (35/83) of dogs and 33.3% (3/9) of cats. Peripheral neurologic system deficits (ataxia, paresis to plegia, absent reflexes, and hypoventilation) were noted in 19.6% (18/92) of dogs and cats. Hemolysis was also common in 37.9% (25/66) of dogs but was not observed in cats. Mechanical ventilation (MV) was indicated in 12% (10/83) of dogs but no cats. Acute kidney injury (AKI), while rare, was a common cause of euthanasia at 20% (2/5) and was the most common complication during MV at 44.4% (4/9). Pigmenturia/hemolysis occurred in 88.9% (8/9) of MV cases and in all cases with AKI. Despite delays in antivenom administration by several hours, dogs and cats with coral snake exposure have low mortality rates (6% of dogs (5/83) and 0% of cats). Gastrointestinal signs were common but were not predictive of progression to neurological signs. Thus, differentiating between coral snake exposure and envenomation before the onset of neurological signs remains challenging.

## 1. Introduction

The eastern coral snake (*Micrurus fulvius*), a member of the elapid family, is an important cause of lower motor neuron paralysis in humans and companion animals in the southeastern United States (U.S.), extending from Florida to North Carolina and Louisiana [1,2,3]. The coral snake has fangs that are poorly effective at delivering venom in a single bite because of its small and fixed dentition (proteroglyphous). Although envenomation can occur with a simple bite, chewing motions can facilitate venom delivery [2]. Therefore, the incidence of dry (non-envenomating) bites can be relatively high [1]. Still, at this time, the incidence of dry bites in small animal medicine is not known.

Generally, eastern coral snake bites cause minimal local tissue reactions, and despite years of experience, it remains impossible to differentiate between exposed and envenomated dogs and cats before the onset of clinical signs. Elapid snake venom consists of different proportions of presynaptic beta neurotoxins (phospholipase-A2) and postsynaptic alpha neurotoxins (3-fingered toxins), depending on the specific genus and species [4]. Eastern coral snake neurotoxin is primarily phospholipase-A2, which blocks nicotinic acetylcholine release at neuromuscular junctions and causes diffuse muscle paralysis of lower motor neurons [5]. Central nervous system signs are rare because of poor venom penetration through the blood–brain barrier [5].

Clinical signs of coral snake envenomation are similar in dogs, cats, and humans, including generalized neuromuscular weakness, ptyalism, vomiting, and respiratory depression [6,7,8]. Respiratory paralysis represents the major cause of death in dogs and humans, and treatment consists of antivenom, mechanical ventilation, and supportive care [8,9]. Dogs are reported to also develop hemolysis due to differences in erythrocyte cell wall composition. The phospholipase-A2 in coral snake venom induces intravascular hemolysis in mice and dog erythrocytes via hydrolysis of the phospholipids in the erythrocyte membrane [8,10]. In contrast, humans, cats, and rabbits resist phospholipase-A2-induced hemolysis due to lower ratios of phosphatidylcholine/sphingomyelin in the erythrocyte plasma membrane [10]. This finding is supported by a retrospective of thirty cats with suspected coral snake envenomation, with only two cats having hemolysis documented, which was expected to be an artifact, supported by the absence of pigmenturia in these cats [11]. However, the incidence of hemolysis in dogs suffering from *Micrurus fulvius* envenomation has not been explicitly reported.

Coral snake antivenom is recommended for all human victims with confirmed bites from *Micrurus fulvius*. If the snake cannot be identified with certainty, it is recommended to hospitalize and observe these patients for 24 to 36 h since clinical signs may not develop for 12 or more hours after the bite [12]. Early administration of antivenom may result in some patients unnecessarily receiving it with the possible risk of hypersensitivity reactions. Symptomatic victims who do not receive antivenom have been successfully managed by providing supportive care, which may include mechanical ventilation (MV) for respiratory paralysis [6].

There are limited data on coral snake envenomation in veterinary patients, especially cats. Current treatment and monitoring recommendations in the United States are largely based on human data and a retrospective study of 20 cases from 1996 to 2011, which proposed a protocol for early antivenom administration to dogs before the onset of clinical signs [8]. While this may have resulted in some dogs receiving antivenom unnecessarily, this has traditionally been accepted as a better outcome than the development of respiratory paralysis and the need for MV. This especially holds true when considering the rising cost of MV veterinary care, which, on average, costs > USD 10,000 and is often cost-prohibitive for many owners. While pet insurance has become more popular in recent years, current estimates report that less than half of pet owners in the U.S have pet insurance, leaving many owners to pay for veterinary care out-of-pocket. Additionally, monitoring recommendations from human medicine and reports of delayed hemolysis in dogs 36 to 72 h after exposure have led to the general recommendation for monitoring veterinary patients for a minimum of 48 h [2,6,8].

The challenges and presumptions required to treat coral snake exposure in veterinary medicine are similar to those in managing snake bites in India in human medicine. The common krait (*Bungarus caeruleus*) in India produces a neurotoxin similar to the eastern coral snake. Similar to coral snake envenomations, the bite is often unwitnessed, and the local tissue reaction of the bite site is small, allowing it to be easily missed, even with a thorough physical exam. Therefore, physicians have to make the presumption of possible envenomation based on clinical signs and the likelihood of exposure [13]. In addition, when clear signs of envenomation are not apparent, but a snake bite is witnessed, “the number of vials of antivenom administered is largely dependent on the experience of the treating physician” [14]. This situation is almost identical to when a dog or cat is presented after coral snake exposure. Unless neurologic signs are present, there is no way to confirm an envenomation nor determine the dose of antivenom needed.

This retrospective observational study assesses the largest cohort of dogs and cats presenting for coral snake exposure in the United States. Although cats compose only a small portion of the reported cases in the veterinary literature, the authors hope that the extra cases will add consistency to what has already been reported previously and provide a direct comparison of species differences between dogs and cats. Focus was given to evaluating clinical signs, time to administration of antivenom, outcomes, and complications. The trends observed from this large sample provide additional information on current management practices, morbidity, and mortality compared to previously published data, where the information provided will allow for standardized management of dogs and cats exposed to coral snakes and set the opportunity for future studies.

## 2. Results

### 2.1. Demographic Information

Ninety-five animals received antivenom for known or suspected eastern coral snake exposure during the study period. Three dogs were excluded for incomplete medical records. Of the remaining 92 cases, 83 were dogs, and nine were cats. Six dogs had a witnessed bite/strike by a coral snake, and sixty-nine dogs were found with a coral snake in their mouth or near a dead coral snake. The remaining eight dogs were treated with coral snake antivenom due to clinical signs consistent with coral snake envenomation and a history of possible exposure based on geographic locale and likely access. All nine cats were found with a coral snake in their mouth or a live or dead coral snake near them. No owners witnessed their cats being bitten.

The mean age was 6.2 ± 3.1 years for dogs and cats. The mean weight for dogs and cats was 26 ± 12.8 kg and 6 ± 1.6 kg, respectively. Among the dogs were thirty-five spayed females, seven intact females, thirty-three neutered males, and eight intact males. Twenty-nine were mixed breeds, and Labrador Retrievers were overrepresented at 8.3% (7/83). Other breeds were present at <5% and are listed in the Appendix A. All cats were neutered male domestic shorthairs. There was no significant association with the development of clinical signs and body weight or signalment.

### 2.2. Clinical Presentation

The most common initial clinical signs in dogs were gastrointestinal at 42.2% (35/83), followed by peripheral neurologic deficits in 20.4% (17/83), and altered mentation (dull to obtunded) in 15.7% (13/83). In cats, gastrointestinal upset was also most common [33.3% (3/9)], followed by altered mentation, and then peripheral neurologic deficits, 22.2% (2/9) and 11.1% (1/9), respectively. Peripheral neurologic deficits included ataxia, weakness, non-ambulatory status, hypoventilation, and decreased spinal reflexes (Table 1). Of these 17 dogs with peripheral neurologic signs, spinal reflex deficits were most common (70.6%, 12/17), while the one cat with peripheral neurologic signs had non-ambulatory tetraparesis. One dog was presented, already sedated and intubated, with manual ventilation provided by the referring veterinary staff, so additional neurologic signs could not be observed in this patient. In addition, two dogs required emergent intubation and initiation of MV on presentation to the university teaching hospital, so testing for other peripheral neurologic deficits was not recorded for these dogs. Of the animals with neurologic signs on presentation, three dogs went on to have progression of paresis/paralysis within the first 12 h of hospitalization. One dog went on to develop hypoventilation, but the remainder of the animals had static to improved neurologic exams throughout hospitalization. No animals that were neurologically normal on presentation developed neurologic signs during hospitalization. Of the 18 animals with peripheral neurologic signs, nine dogs and one cat had concurrent mental depression (55.6%, 10/18). Of the 21 dogs with neurologic signs (peripheral deficits or mental depression), 57.1% had concurrent gastrointestinal signs (12/21), but only 34.3% of dogs with gastrointestinal signs had concurrent neurologic signs (12/35). No cats had an overlap of gastrointestinal and neurologic signs.

Definitive evidence of bite wounds was rarely reported, but seven dogs (8.4%) were described to have dried blood/areas of pinpoint, mild swelling, or pain around the muzzle/lips (Figure 1). No wounds or local tissue reactions were noted in cats.

Packed cell volume and plasma total solids (PCV/TSs) were obtained in 66 dogs, and hemolysis was noted in 37.9% (25/66). At the same time, pigmenturia was only reported in 14.4% (12/83) of dogs based on history, physical exam, or observations during hospitalization. However, only five dogs had urinalysis performed, three of which had gross pigmenturia (Figure 2). Of the 25 dogs with hemolysis, 56% had concurrent neurologic signs (14/25), and of 12 dogs with pigmenturia, 75% had concurrent neurologic signs (9/12). All acute kidney injury (AKI) cases were noted to have hemolysis and gross pigmenturia and were undergoing MV, while AKI was not documented in any dogs that were not ventilated. It was impossible from the medical records to compare the severity of hemolysis and pigmenturia between dogs that developed AKI and those that did not. In contrast, no cats had documented hemolysis in the seven cats that had a PCV/TS performed. In addition, no AKI or pigmenturia was noted in any cat.

Two dogs developed significant anemia secondary to hemolysis within the first 24 h of hospitalization (dog 1’s PCV dropped from 55% to 14%, and dog 2’s PCV dropped from 38% to 15%). Only the first dog required treatment with a packed red blood cell transfusion. This dog had hemolysis, pigmenturia, and spherocytosis noted initially. The dog was given a second vial of antivenom after the severe anemia was noted on day 2 and was hospitalized for five days until resolution of both hemolysis and pigmenturia was achieved. This dog never developed any neurological signs.

Blood smears were performed for 48 animals, and echinocytes were noted in 31.3% of the cases (15/48, 14 dogs, and one cat). Quantifying the estimated severity of echinocytes from this retrospective data set was not possible due to the lack of a consistent scoring system. Most cases used descriptors such as “scant”, “few”, “occasional”, “rare”, or “1+” to enumerate the echinocytes. Only one dog had “too numerous to count echinocytes”. For the 78 animals with a blood gas analysis performed, hyperlactatemia (defined as lactate > 2.5 mmol/L) was the most common abnormality occurring in 30.8% (24/78) with a mean lactate of 3.78 mmol/L in animals for which hyperlactatemia was noted.

Mechanical ventilation was indicated in 12.0% (10/83) and performed in 10.8% (9/83) of dogs due to one owner declining care due to financial constraints. The decision to implement MV was based on a subjective assessment of detecting shallow chest excursions and detecting hypercapnia (PCO_2_ > 45 mmHg) on the blood gas analysis. Hypercapnia was documented in five of these dogs, with a mean PCO_2_ of 50.8 mmHg. Four other dogs had apparent shallow inspiratory chest movements upon physical examination. A referral veterinary hospital staff member transferred the last dog while being ventilated due to the dog already undergoing respiratory paralysis while at their facility. MV was not performed in any cats. Of the dogs receiving MV, the decision to ventilate was made within the first 4 h of hospitalization, except for one dog that initially presented obtunded with tetraparesis and pigmenturia. This dog was initially determined to have adequate ventilation (initial PCO_2_ 34.5 mmHg), but then was noted to have shallow respirations and mild hypercapnia (PCO_2_ 47 mmHg) approximately twelve hours after presentation when MV was initiated. Of the dogs receiving MV, 66.6% survived to discharge with a complete recovery. The mean duration on the ventilator was four days. The mean total visit cost for cases undergoing MV was USD 10,570 (range, USD 8136.26–USD 17,812). Complications associated with MV were common at 66.7% (6/9) and included AKI 44.4% (4/9), corneal ulceration 22.2% (2/9), and pneumonia 22.2% (2/9). No incidence of barotrauma was noted. Hemolytic and gastrointestinal signs were also common among dogs undergoing MV, occurring at 88.9% (8/9) and 44.4% (4/9), respectively (Table 2). The typical ventilator setup for a dog suffering from respiratory paralysis is depicted in Figure 3.

### 2.3. Antivenom Administration & Monitoring

The median number of administered antivenom vials was similar at 1.31 (1–4) for dogs and 1.22 (1–2) for cats. Administering more than two vials only occurred in three dogs. The first dog received four vials and was placed on the ventilator on presentation due to severe neurologic dysfunction and respiratory failure. The second dog received three vials due to persistent hemolysis and pigmenturia. The reason why the third dog received three vials on presentation, while only showing ptyalism and no neurologic signs, was not explicitly provided. The mean time from presentation to coral snake antivenom administration was 2.26 ± 1.46 h. Mean time to antivenom administration was longer in dogs than cats [2.29 (0.42 to 10.67) vs. 1.95 (0.75 to 6.13) hours from presentation, respectively]. There was one outlier in the dog group, where the time from presentation to administration was >10 h because of a shipment delay for the antivenom. The level of clinician experience had no significant effect on the number of vials administered (*p* = 0.459). While the time to antivenom administration was shortest for residents, this difference approached but did not reach statistical significance (*p* = 0.05, Table 3). The number of cases with progressive clinical signs was too small to compare disease progression with time to antivenom administration.

Immediate adverse reactions to coral snake antivenom administration were reported in three dogs (3.6%, 3/83) and no cats. One dog vomited and developed tachycardia 10 min into the transfusion, which was administered over 30 min. The dog was given maropitant (1 mg/kg IV), and the transfusion was slowed to give the remainder for over 2 h. The dog’s tachycardia resolved with no further signs of an adverse reaction. Two other dogs developed anaphylaxis characterized by urticaria and vomiting and collapsed within 5 min of starting antivenom administration. Both dogs were treated with an IV fluid bolus, diphenhydramine [2 mg/kg intramuscularly (IM)], and epinephrine (0.01 mg/kg IM, repeated once, followed by a constant rate infusion of epinephrine at 0.01 to 1 mcg/kg/min (titrated to effect to maintain a systolic blood pressure > 100 mmHg). One of these two dogs had received coral snake antivenom the year prior. Both dogs remained on telemetry and frequent blood pressure monitoring overnight. The following morning, both dogs had normal vital signs, and the epinephrine infusions were discontinued. Neither of these two dogs completed their antivenom infusion, but fortunately, they did not develop any signs of neurologic dysfunction or hemolysis. Although follow-up was not documented on every patient after discharge, any additional recorded correspondence (most commonly a telephone call several days after discharge) revealed no indication of delayed adverse reactions at home.

Twelve dogs and one cat were given antivenom and subsequently monitored in the emergency room for eight hours due to owners declining hospitalization (primarily due to financial constraints). These patients were then sent home for further monitoring for the next three days, and no adverse effects were reported in these animals. After excluding these animals, the mean length of hospitalization for dogs and cats was 50.8 h and 34 h, respectively. When looking specifically at dogs requiring MV, the mean length of hospitalization was increased to 7.5 days. When excluding dogs requiring MV, only four dogs were hospitalized for ≥5 days. All of these dogs developed non-ambulatory tetraparesis; three dogs had hemolysis noted, and two had hypercapnia on blood gas. Still, they were determined not to have clinical hypoventilation sufficient to require MV, and two dogs suffered from aspiration pneumonia. All these signs were resolved at the time of discharge for all four dogs.

### 2.4. Outcomes

The mortality rate for dogs was 6.0% (5/83), while all cats survived to discharge. Of the dogs that died, one dog spontaneously died due to cardiac arrest shortly after presentation, and one dog died at home after the owner declined MV and euthanasia. The remaining three dogs were euthanized, and all of these dogs received MV. Two of these dogs were euthanized due to the development of progressive AKI while on the ventilator. The third dog was discharged with improved clinical signs regarding coral snake envenomation after receiving multiple rounds of CPR and being mechanically ventilated. During the hospitalization of this third dog, extravasation occurred in a hind limb peripheral IV catheter, resulting in the accumulation of potassium chloride and metoclopramide in the soft tissues of the hindlimb. Following this third patient’s discharge, correspondence with another hospital confirmed the affected limb progressed to severe soft tissue necrosis, providing the basis for euthanasia. Too few cases died to assess the time of antivenom administration effect on outcome.

## 3. Discussion

This report provides the most extensive study on eastern coral snake exposure in the United States and associated antivenom administration in cats and dogs. All 93 of these coral snake encounters occurred in Florida, with each month of the year represented at least once, supporting the premise that coral snake exposure occurs year-round in this particular geographic location. With almost 100 cases in two years, this study documents a significant increase in reporting of coral snake exposure in small animals, which contrasts with previous veterinary case reports from the same Florida institution that documented only 20 cases over 15 years [8]. These changes may be explained by the increases in pet population, general awareness of coral snake toxicity, referral by primary veterinarians, motivation to seek veterinary care by owners, and expansion of urban development into natural habitats of coral snakes.

This study found that upper gastrointestinal signs (ptyalism and vomiting) were commonly reported at 41.3% of dogs and cats exposed to coral snakes (38/92). The incidence of gastrointestinal signs reported here is greater than the 20% reported in a 20-case series from 1996 to 2011 but similar to a smaller study of eight animals undergoing MV after coral snake envenomation in which 50% (4/8) initially presented with gastrointestinal signs [8,9]. However, in this study only 34.3% (12/35) of dogs with gastrointestinal signs had concurrent neurologic signs, and no cats had an overlap of gastrointestinal and neurologic signs. Therefore, gastrointestinal signs do not necessarily indicate that an envenomating bite has occurred. *Micrurus fulvius* has been shown to “musk”, which is the process of releasing foul-smelling, nauseating discharge from their cloaca as part of their antipredator display [15,16]. This musk comes in contact with the oral mucosa of small animals when they bite a coral snake and is a likely explanation for gastrointestinal signs in animals that do not go on to develop neurologic signs. It, therefore, remains impossible to predict a patient’s progression to neurological deficits [2]. Conversely, 57.1% (12/21) of dogs with neurologic deficits had preceding or accompanying gastrointestinal signs, which suggests that clinicians should be prepared to treat gastrointestinal signs in dogs with neurologic deficits.

In this study population, the onset and progression of neurologic signs were highly variable, consistent with previous reports [8]. Excluding dogs requiring MV, only four dogs required prolonged hospitalization for tetraparesis, and no dogs with normal neurologic exam on presentation developed hypoventilation or other neurologic signs. The mean time to develop hypercapnia was approximately five hours after presentation with a range of three to 12 h, and no dogs were placed on MV after 12 h of monitoring. Therefore, outpatient treatment with prophylactic antivenom administration for asymptomatic patients or shorter periods of in-hospital monitoring for hypoventilation may be considered.

This is also the first study to explicitly describe the incidence of hemolytic signs in dogs exposed to *Micrurus fulvius*. Like gastrointestinal signs, hemolysis was common at 37.9% but failed to show a strong association with the development of neurologic signs, with 56% of dogs with hemolysis having concurrent neurologic deficits. This study is also the first to report a dog developing coral snake envenomated-induced anemia severe enough to require treatment with a blood transfusion. Therefore, clinicians need to be aware that severe hemolysis is a possible, albeit rare, complication of coral snake envenomation in dogs. Pigmenturia, while the least common sign at 14% (12/83), may have the strongest correlation to neurologic signs in dogs. While historically, pigmenturia has been reported as a delayed clinical sign, it was documented during presentation in nine of the 12 dogs. Furthermore, 75% of dogs with pigmenturia had concurrent neurological signs. Therefore, pigmenturia may have the potential to be a screening tool for envenomation. This supports the need to make a urinalysis a part of the minimum initial diagnostic database for coral snake exposure in dogs. Future studies comparing the time to development of pigmenturia to the time of onset of neurologic signs would be needed to confirm the utility of this correlation in dogs.

As most coral snake exposures (bites and direct contact with the patient) were not confirmed with 100% owner certainty, no apparent “magic window” of time for antivenom administration can be accurately calculated, especially because lymphatic venom absorption from the bite site might require several hours of delay [17,18]. Previous recommendations in both veterinary and human medicine encourage early antivenom administration even without clinical signs, as life-threatening neurologic signs can have a delayed onset of 12 to 24 h [6,8,9]. The clinicians where this study took place have adopted similar guidelines for antivenom administration for all patients with known or highly suspected exposure to an eastern coral snake [8,9]. The average number of antivenom vials administered to these patients was 1.29, with the average administration occurring within two and one-half hours. There is no direct correlation between clinician experience and the timing or number of vials of antivenom treatment. General protocols for initiating coral snake antivenom at this institution call for one to two vials promptly upon presentation. This stems from a pragmatic approach of spending USD 400–800 on antivenom to hopefully prevent progression to respiratory paralysis requiring MV with average costs of >USD 10,000. The number of vials received and the timing of administration are also largely influenced by the pet owner’s financial constraints. Some delays are due to the time required for owners to consent to treatment in the setting of a high-case-load emergency room. Assuming that the antivenom product contains adequate amounts of neutralizing antibodies against the harmful venom components, the dose administered would be based on the average amount of venom injected by an eastern coral snake bite (2–20 mg), which is one to three vials for the particular antivenom used in this study according to manufacturer guidelines [19].

Given the low incidence of progressive neurologic signs in the study population, the role of antivenom in preventing disease progression and hypoventilation remains unclear. One dog that became dull within 45 min of exposure received two vials of antivenom within two hours of admission. This dog became progressively weaker upon rising during the first night in hospital, so it was given an additional two vials of antivenom at 19 h post-exposure. Fortunately, this dog’s paresis improved, and it was discharged after 44 h in the hospital with no development of hypoventilation, suggesting a protective effect of antivenom. Another dog that progressed to a non-ambulatory state during a four-hour commute to the hospital received one vial of antivenom within an hour of presentation and an additional vial three hours later. However, after 12 h, the dog progressed from tetraparesis to flaccid paralysis with mild hypercapnia (PCO_2_ 47 mmHg). Fortunately, this dog did not require MV but was hospitalized for 96 h with more extensive recumbency care compared to the dog treated earlier after exposure and received additional antivenom. The variability in antivenom administration with predominantly positive outcomes raises the questions of how much antivenom is truly needed to neutralize eastern coral snake venom, when and if antivenom administration should be repeated, and how effective is the body’s natural clearance system for coral snake venom with appropriate supportive care, with the consistent unknown variable being the amount of venom injected by the snake.

Adverse effects secondary to antivenom administration are common in people with allergic reactions, being as high as 18.5%, and anaphylaxis as high as 8% with certain types of antivenom, while anaphylaxis secondary to coral snake antivenom has been previously reported in one dog [1,8,20,21]. However, there are no previous reports in veterinary medicine of sensitization with prior antivenom exposure, as was suspected in one of the study dogs that was administered antivenom a year prior. A similar event was reported in a human suffering anaphylaxis after receiving antivenom a second time after being bitten by a snake one month prior [22]. From this report, it is evident that antivenom exposure can produce sensitization through the production of immunoglobin E (IgE) antibodies, leading to a Type I hypersensitivity reaction upon re-exposure of the offending antigen, while understanding that a Type 1 reaction can also occur in the absence of IgE [23]. While treatment with emergency epinephrine (0.01 mg/kg IM) has been previously reported for the treatment of anaphylaxis secondary to antivenom administration, the pre-drawing of epinephrine during antivenom administration was not routinely practiced in this patient population, presumably due to the low occurrence of such reactions (<5%) [8]. This lack of hypersensitivity reaction agrees with previous veterinary studies that have documented a low incidence of adverse reactions with coral snake antivenom administration [8,9,11]. However, caution should be exercised regarding the possibility of such reactions.

Acute kidney injury was the most common complication in MV patients (44.4%, 4/9) and the most common reason for euthanasia in this study population (20%, 2/5). In a previous evaluation of MV in seven coral snake envenomated dogs, two dogs developed AKI, and one of these dogs was euthanized [9]. While AKI is a known complication of MV, studies reporting the incidence of AKI in dogs undergoing MV are unknown [24]. In humans, the incidence of ventilator-induced kidney injury has been reported to be as high as 45% [25]. Previous studies in anesthetized dogs undergoing MV have documented variable changes in renal perfusion and increased excretion of antidiuretic hormone as possible explanations for ventilator-induced renal injury [26,27], which might explain the onset of AKI in this study population. Pigmenturia can induce oxidative renal injury [28] and likely also contribute to AKI, as all dogs that developed AKI had documented pigmenturia. Furthermore, one of these dogs with pigmenturia on presentation developed azotemia prior to initiation of MV, suggesting that the AKI, in this case, was likely induced by pigmenturia secondary to envenomation and possibly worsened by subsequent ventilation. However, the incidence of AKI with coral snake envenomation was low, occurring in only 4.8% (4/83) of this study population. While the exact mechanism of coral snake envenomation-induced AKI remains poorly understood, serial renal function assessments should be considered in those patients that develop pigmenturia.

This study shows that the prognosis of coral snake exposure after antivenom administration was excellent, with survival in 100% of cats and 94% of dogs. Besides the one dog that was euthanized four days after discharge related to catheter site infection, no owner reported complications after discharge. These findings are consistent with the previous literature reporting 87.5 to 100% survival in cats and dogs [8,9,11]. The patient mortality rate of ventilator patients was 33% in this study, while the mortality rate in a previous study was only 12.5% [9]. The average time to weaning MV in this study was also longer at 3.6 days compared to 2.4 days. These differences may be explained by delays in treatment decisions, differences in coral snake inoculums, or the fact that the current study’s patient population was more severely affected by complications such as hemolysis and AKI.

Although the numbers are low, this study describes the second-largest reported sample of cats with coral snake exposure in the U.S. and provides a valuable comparison of how clinical manifestations can differ in cats from dogs. Of the nine cats in this study, only one had peripheral neurologic deficits, and all cats survived, with none requiring mechanical ventilation. Pigmenturia was not documented in any cats because of the known absence of coral snake-induced hemolysis in this species, thereby causing clinicians not to assess the urine. These findings agree with a previous retrospective of 30 cats exposed to coral snakes from 2012 to 2019, which reported no significant hemolysis and a 97% survival rate [11]. Together, these findings support that cats can have an excellent prognosis after coral snake exposure and that feline erythrocytes are more resistant to the hemolytic effects of phospholipase-A2 in coral snake venom compared to dog erythrocytes [10]. As previously described, this resistance is likely a result of a lower ratio of phosphatidylcholine/sphingomyelin in the erythrocyte plasma membrane compared to dogs [10]. In addition, no AKI was reported in cats in the present and the previous study, which further implicates the role of hemolysis and pigmenturia in the development of AKI in susceptible species such as dogs.

Lastly, this highlights the need for a point-of-care diagnostic test to differentiate coral snake envenomation from simple exposure, as less than one-third of the animals progressed to develop neurologic signs (25.3% of dogs and 22.2% of cats). Previous studies have called for point-of-care diagnostic tests specific to *Micrurus fulvius* venom. An ELISA test for coral snake venom has been previously described in people, and there is ongoing research on a multiplex lateral flow assay to differentiate different venomous snakes in Brazil [11,29,30]. This would allow for significant savings in resources (antivenom, cage space for patients under monitoring for development of clinical signs, nursing care, etc.) and reduce costs to owners. Furthermore, a definitive diagnosis of envenomation would allow for more direct conversations with owners about the expected progression of clinical signs and stronger justification of antivenom use and inpatient monitoring protocols. Once a diagnostic test for envenomation has been developed, future projects aimed at measuring serum concentrations of venom on presentation, as well as the amount present after antivenom administration, could help optimize antivenom protocols for coral snake envenomation in canine and feline patients.

### Limitations

The limitations of this study include its retrospective design and the lack of corroborative diagnostics available to veterinarians in practice. As most of the cases were not directly witnessed being bitten by a coral snake, in addition to those that might have been dry bites, it is impossible to ascertain how many of these patients may have been bitten without an ability to measure venom antigen, thus hampering any statistical evaluation. The inclusion criteria for this study only selected patients billed for coral snake antivenom, suggesting that there may have been an unreported population of coral snake-exposed patients not treated with coral snake antivenom (which does occur routinely at our institution due to owners declining treatment for financial reasons). Further limitations of this study are the lack of standard diagnostic and treatment protocols. For example, urinalysis should be included in the minimal database on dogs exposed to coral snakes to assess the true incidence of pigmenturia after envenomation. Pigmenturia was documented in only 14.5% of this study population, while hemolysis was documented in 37.9% of dogs that had a PCV/TS performed. These results suggest that only a small portion of dogs with coral snake venom-induced hemolysis develop pigmenturia; however, it is highly likely that urine discoloration may have been missed by hospital personnel or that microscopic pigmenuria, only detectable on urinalysis, was missed upon gross examination. Further evaluation of the true incidence of pigmenturia in coral-snake-exposed dogs becomes especially important when considering that 75% of dogs with pigmenturia had neurologic signs and 88.9% of the dogs on MV had concurrent pigmenturia.

In addition, a larger population of cats would have allowed for a stronger comparison of species differences. This study also failed to properly compare the severity of echinocytosis in dogs due to inconsistent reporting of echinocyte number and echinocytes, which were not adequately screened for in cats, with only three of nine cats having a blood smear performed. In contrast, the previous study reported echinocytes in 85% of cats that showed clinical signs of envenomation [11]. Future prospective studies with standard diagnostic, treatment, and monitoring protocols, which are not limited by owner finances, may provide a more comprehensive understanding of the optimal management of small animal coral snake victims.

## 4. Conclusions

Given that previous recommendations for coral snake exposure in veterinary medicine are based on similar retrospective studies, the authors have developed the following updated recommendations: Treatment with antivenom remains recommended for any dog or cat with coral snake exposure. A minimum of one vial of coral snake antivenom should be administered over 30 min to one hour, depending on the animal’s clinical status, or the animal should be referred to the nearest hospital with access to antivenom and the capability to perform mechanical ventilation, if indicated. Gastrointestinal signs are common but may be related to coral snake musk and do not indicate envenomation. The development of lower motor neuron (LMN) neurologic deficits or hypoventilation remains the strongest indicator of envenomation after coral snake exposure. The combination of LMN and hemolytic signs in dogs may help distinguish coral snake envenomation from other causes of LMN disease, such as tick paralysis, myasthenia gravis, botulism, and polyradiculoneuritis. However, envenomated cats may be more challenging to diagnose due to their intrinsic resistance to hemolysis. This highlights the need for a point-of-care diagnostic test to confirm coral snake envenomation. Baseline diagnostics should include a venous blood gas to screen for hypercapnia and hyperlactatemia and a blood smear to screen for echinocytes. PCV/TS and urinalysis should also be performed in dogs to screen for hemolysis and pigmenturia. In-hospital monitoring is still recommended for a minimum of eight hours for asymptomatic patients and 12 h for patients with any neurologic signs to document disease progression and provide necessary interventions. However, outpatient protocols after antivenom administration are also reasonable for asymptomatic animals. If mechanical ventilation is performed for a coral snake victim, all the standard complications associated with MV should be considered, with special emphasis on AKI in dogs with concurrent pigmenturia. Despite these complications and delays in antivenom administration by several hours, dogs and cats with coral snake exposure have low mortality rates.

## 5. Materials and Methods

The medical records of dogs and cats examined and hospitalized at a College of Veterinary Medicine teaching hospital, which manages primary emergencies through tertiary referrals, between January 2021 and November 2022, were searched electronically for invoice items containing coral snake antivenom. Animals were excluded from the study if the medical records were incomplete. By searching for the invoice charge for antivenom, dogs and cats presented for coral snake exposure but did not receive antivenom were not captured in this retrospective search. This search method was utilized due to the lack of consistent diagnostic code for coral snake exposure and due to the limitations of our electronic medical record database search functions.

Data collected included signalment, whether or not the bite was witnessed, initial clinical signs, physical examination, initial diagnostics, the number of vials of antivenom administered, and time from presentation to antivenom order. Clinical signs were divided into gastrointestinal (vomiting and ptyalism), neurologic (decreased mentation, ataxia, flaccid paralysis, loss of spinal reflexes, and hypoventilation), and hemolytic (hemolysis and pigmenturia). Given the retrospective nature of this study, no set diagnostic tests were carried out uniformly for all cases. Initial packed cell volume and total solids (PCV/TSs), venous blood gas (NOVA Stat Profile®, Nova Biomedical Corp, Waltham, MA, USA), blood smear, and urinalysis were reviewed in all cases for which these diagnostic tests were performed. The duration of hospitalization, use of MV (Puritan Bennet 840, Medtronic, Minneapolis, MN, USA). and survival to discharge were also recorded. Identifying clinician experience was also recorded and broken down into three groups: intern, resident, and faculty.

For patients undergoing MV, additional parameters were recorded, including time weaning from MV, complications, and visit cost. MV complications were defined as follows: acute kidney injury (AKI), documented by a rise in serum creatinine by >0.3 mg/dL within 48 h; corneal ulcers, documented by a positive fluorescein stain uptake; pneumonia, documented by a new radiographic pulmonary infiltrative pattern with concurrent fever or increase in white blood cell count; and barotrauma, documented with pneumothorax on chest radiographs or via thoracocentesis during or within 24 h from weaning off MV.

All patients received the same type of antivenom, Anti-Coral (Instituto Clodomiro Picado, San Jose, Costa Rica), a liquid IgG equine-derived antivenom that is imported from Costa Rica with a special USDA permit. This product is composed of antibodies to several coral snake species, including 1.25 mg from *M. fulvius*, 3.0 mg from *M. nigrocinctus*, and 3.0 mg from *M. carinocaudus*. In light of the cross-neutralization of venom between species, the recommended dose for neutralizing the average coral snake envenomation is one to three vials, according to manufacturer guidelines. Antivenom was diluted with 50–100 mL 0.9% sterile saline and administered intravenously over 15–60 min.

Descriptive data are presented as mean ± S.D. Statistical analyses were performed using commercial software (Excel 2016, Microsoft Corp, Redmond, WA, USA). To assess the effect of clinician experience level of number of antivenom vials administered and time to antivenom administration, ANOVA was run on ranked data with a *p*-value of <0.05 considered significant.

## Figures and Tables

**Figure 1 toxins-16-00246-f001:**
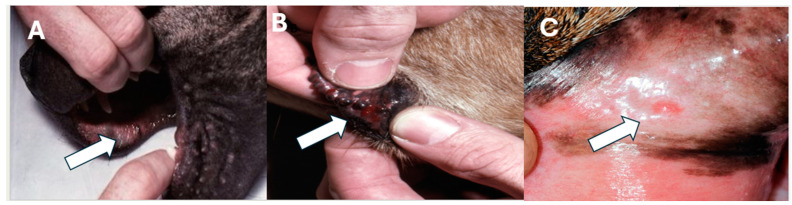
Three examples of the pinpoint tissue reaction left by the small fangs (proteroglyphous) of the eastern coral snake in dogs. (**A**) Small bleb on the oral mucosa of the upper lip, (**B**) slit-like wound on the outer lower lip, and (**C**) focal erythematous area on gingiva. (Courtesy, M. Schaer, University of Florida).

**Figure 2 toxins-16-00246-f002:**
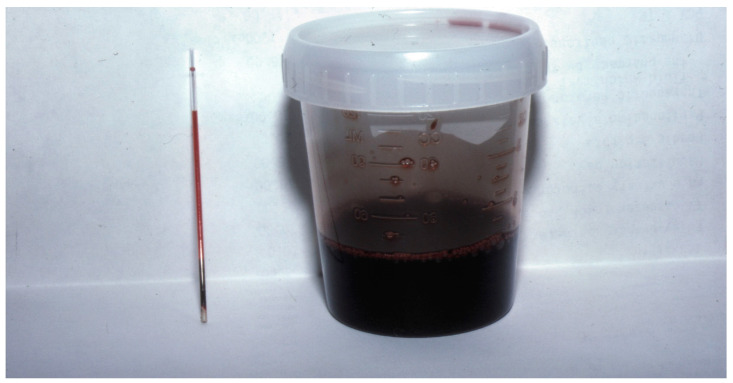
An example of hemolysis (in the capillary tube on the **left**) and pigmenturia (in the urine collection cup on the **right**) from a dog undergoing hemolysis caused by eastern coral snake envenomation. This dog also had lower motor neuron dysfunction. Courtesy: M. Schaer, University of Florida.

**Figure 3 toxins-16-00246-f003:**
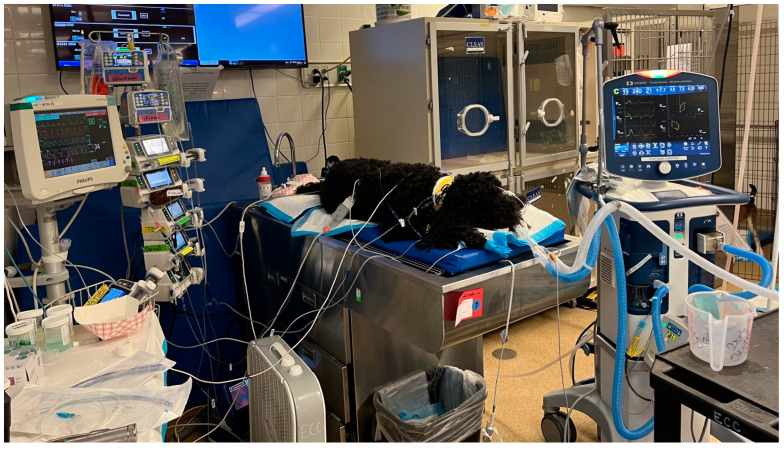
A mixed-breed dog that was envenomated by an eastern coral snake and had progressive lower motor neuron dysfunction with eventual respiratory paralysis that required ventilatory support for three days. Courtesy: Dr. M. Schaer, University of Florida.

**Table 1 toxins-16-00246-t001:** Clinical signs after reported coral snake exposure in dogs and cats.

Clinical Signs	Dogs (*n* = 83)	Cats (*n* = 9)
Ptyalism	23 (27.7%)	1 (11.1%)
Vomiting	15 (18.1%)	3 (33.3%)
Dull to obtunded mentation	13 (15.7%)	2 (22.2%)
Ataxia	7 (8.43%)	0
Decrease or loss of spinal reflexes	12(14.5%)	0
Non-ambulatory paresis/paralysis	9 (10.8%)	1 (11.1%)
Hypoventilation	11 (13.3%)	0
Tremors/seizures	2 (24.1%)	0
Hemolysis	25 (37.8%); *n* = 66	0
Pigmenturia	12 (14.4%)	0

**Table 2 toxins-16-00246-t002:** Clinical signs (gastrointestinal, mentation change, peripheral neurologic, and hemolytic; presence denoted by an “X”, duration of mechanical ventilation (MV) in days, duration of hospitalization in days, complications (corneal ulcers = C.U.; acute kidney injury = AKI; bacterial pneumonia = B.P.; and other), visit cost in U.S. dollars (USD), and outcome (survived vs. euthanasia) for the nine dogs undergoing MV.

	Dog 1	Dog 2	Dog 3	Dog 4	Dog 5	Dog 6	Dog 7	Dog 8	Dog 9
Initial PCO_2_ (mmHg)	74.8	55.4	46.8	72.1	52.9	30.1	39.6	34.5	n/a
Gastrointestinal signs		X	X		X		X		
Dull/obtunded mentation	X	X	X			X	X	X	X
Peripheral neurologic signs	X	X	§	§	X	X	§	X	X
Hemolysis/pigmenturia	X	X	X	X		X	X	X	X
Number of antivenom vials	4	1	2	2	2	2	1	2	2
Duration of MV (days)	2	4	4	3	3	1	7	4	4
Duration of hospitalization (days)	8	8	7	3	9	1	16	10	11
Ventilator-associated complications	None	None	C.U. Other	AKI	None	AKI	AKI C.U. B.P.	AKI	B.P.
Visit cost (USD)	8136	11,322	9849	7936	10,990	2617	17,812	13,926	12,538
Outcome (survived = 1; euthanized = 2)	1	1	2 *	2	1	2	1	1	1

§ Peripheral neurologic signs were not noted as these dogs required immediate MV on presentation. “Other” complications in dog #3 include multiple episodes of cardiopulmonary arrest, urinary tract infection, and extravasation of intravenous fluids, progressing to necrotizing fasciitis. * Although dog #3 survived to discharge, it was euthanized four days later, secondary to these complications.

**Table 3 toxins-16-00246-t003:** The effect of clinical experience level (intern, resident, and faculty) on time to antivenom administration in hours and the total number of vials administered to dogs and cats with coral snake exposure. Data are presented as mean ± S.D.

Clinician Experience Level	Time to Antivenom (h)	Number of Antivenom Vials
Interns (*n* = 31)	2.49 ± 1.94	1.23 ± 0.57
Residents (*n* = 41)	1.94 ± 1.32	1.32 ± 0.47
Faculty (*n* = 21)	2.56 ± 1.10	1.38 ± 0.74

## Data Availability

Data available at https://uflorida-my.sharepoint.com/:x:/g/personal/jsullivan18_ufl_edu/ESzPsswyQidFhgctqMFl0McBclpIEaNztTPnOZO74CN2FA?e=VNpk2H (accessed on 5 March 2024).

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
