# Peer review of "Retrospective Evaluation of Clinical and Clinicopathologic Findings, Case Management, and Outcome for Dogs and Cats Exposed to Micrurus fulvius (Eastern Coral Snake): 92 Cases (2021–2022)"

_toxins, 2024, doi:10.3390/toxins16060246_

Round 1

Reviewer 1 Report

Comments and Suggestions for Authors

The manuscript is well written, and their findings are important to clinicians. Nevertheless, some concerns must bu solved before its publication. 

Abstract

Line 6. Retrospective…… Clarify, retrospective observational study

Intriduction

Micrurus fulvius must be in italics. Review all document.

Line 44 and 55. It must be Phospholipases A2. It is important to review this in all document, because, snake venoms contains other type of Phosphilipases, such as B.

Line 46. It must be phospholipases A2.

Line 46-48. Authors should differentiate the mode of action of both group of toxins, 3FTxs and PLA2s. The first group blocks ACh receptors, but the second group blocks the realease of ACh. In addition, this paragraph lacks references.

Materials and methods

Line 52 include as an exclusion criterion that animals were euthanatized before antivenom treatment. Nevertheless, in the results section there is any information about this topic. Authors must clarify how many animals were euthanatized before antivenom administration and their causes to make this decision.

Results

Results are well described. I recommend including the number of animals that were animals were euthanatized before antivenom administration and specify if the cause was related to coral envenomation.

Discussion

Line 264. Include reference for the previous report of the same institution.

Line 328. Change this review for tis study,

Paragraph in lines 511 to 541 must be include as the occlusion of this study.

In addition, I recommend to mention the need to standardized the presentation of sings, symptoms and laboratory results un medical records. Discuss this for echinocytes. Another important finding that was not discussed was the hyperlactemia.

Finally, the changes in the blood cell membrane in cats may explain the ir resistance to hemolysis and further AKI, nevertheless, this fact do not explain their resistance to neuromuscular effects of coral venom. Therefore, authors mus disscuss this topic. N of this studty can be ralated?

Reviewer 2 Report

Comments and Suggestions for Authors

Well done to the authors to bring together this case series on a little documented snake envenomation in animals.  I am surprised at the number of cases seen in a relatively short time period.  This paper needs to little work to tighten up the discussion.

Results

Any thoughts on why cats don’t seem to get haemolysis in contrast to dogs?

Any relationship between severity and dog bodyweight?

MM

Line 568           Please include details of the university teaching hospital.

Discussion

Lines 256-556              The discussion is rather long and occupies 50% of the paper.  This should be trimmed markedly and made more precise.

Line 351                          Onset of signs is likely related to venom dose/kg of bodyweight. Lewis, P. F. (1994). "Some toxicity thresholds for the clinical effects of common Tiger Snake (Notechis scutatus) envenomation in the dog." Australian Veterinary Journal 71(5): 133-135.

Line 476                          Change “...bit...” to “...bitten…”

Round 2

Reviewer 1 Report

Comments and Suggestions for Authors

The authors paid attention to the major concerns. At this moment, the manuscript is clearer than the first submitted document.